

# Frequency anomalies and characteristics of extratropical cyclones during extremely wet, dry, windy and calm seasons in the extratropics

Hanin Binder[1] and Heini Wernli[1]

[1]Institute for Atmospheric and Climate Science, ETH Zurich, 8092 Zurich, Switzerland

**Correspondence:** Hanin Binder (hanin.binder@env.ethz.ch)

**Abstract.**

Extreme meteorological seasons are highly relevant because of their severe impacts on many socioeconomic sectors. However, a global statistical characterisation of observed extreme seasons is challenging, because at any specific location only very few such seasons occurred during the limited period with available reanalysis data sets. This study therefore uses 1050 years of present-day (1990-1999) climate simulations of the Community Earth System Model Large Ensemble (CESM-LE) to systematically identify extremely wet, dry, windy and calm seasons in the Northern Hemisphere (NH) and Southern Hemisphere (SH) extratropics during winter and summer, and to quantify the role of extratropical cyclones for their occurrence. Extreme seasons are defined as spatially coherent regions of extreme seasonal mean precipitation or near-surface wind. Compared to the climatology, extremely wet seasons are associated with positive anomalies in cyclone frequency in large parts of the extratropics. In the SH storm track and at the downstream ends of the NH storm tracks, cyclones contributing to wet winters are also anomalously intense and typically originate unusually far to the west and south, while in the subtropical North Atlantic and over the eastern Mediterranean they are on average more stationary than climatologically. During wet summers, many continental regions are not associated with strong anomalies in any of the cyclone characteristics (e.g., most of North America, the coastal regions around the Mediterranean Sea and southern Asia), which points to the importance of other processes like convection, orographic ascent or, over southern Asia, monsoon precipitation. Windy seasons are often associated with anomalously few, but particularly intense cyclones, especially during winter. Positive anomalies in both cyclone frequency and intensity are found in the southern North Atlantic during winter, which suggests that windy winters in this region occur during southward shifts in the position of the main storm track. The patterns of dry and calm seasons mainly contrast with those of wet and windy seasons, i.e., they are often characterised by particularly few or weak cyclones or a combination thereof. In all four types of extreme seasons, there is remarkably large spatial and seasonal variability in the cyclone properties, especially over the continents. In addition to the systematic analysis based on the climate model, two past extreme seasons have been studied with ERA5 reanalyses over the United Kingdom, the wet winter 2013/14 associated with anomalously many and intense cyclones and the windy winter 1988/89 associated with anomalously few, but intense cyclones. The results are consistent with those from the climate model for this region, suggesting that the model captures the cyclone properties reasonably well during extreme seasons. Overall, it can be concluded that (i) anomalies in the seasonal frequency and/or intensity distribution




of extratropical cyclones are crucial for the occurrence of many extreme seasons in the extratropics, and (ii) this link shows substantial geographical and seasonal variability.

# 1 Introduction

Meteorological extreme events on the seasonal timescale can have severe social, environmental and economic impacts. For instance, in winter 2013/14 the United Kingdom (UK) and Ireland experienced extremely wet and windy conditions, which resulted in widespread flooding and damage to forests, infrastructure, transportation networks and power supplies (Davies, 2015; Kendon and McCarthy, 2015). Another example is the dry and hot European summer 2003, which was associated with increased mortality rates, reduced crop yields, forest damage and increased prices for electricity (Fink et al., 2004). The integrated weather conditions over an entire season can negatively affect many socioeconomic sectors like public health, agriculture, freshwater supply, energy production, tourism and the financial sector. Therefore, understanding the physical factors driving extreme meteorological seasons is of high relevance.

In contrast to seasonal extremes, isolated extreme events on timescales of a few minutes to several days have been studied extensively during the last decades. Single extreme precipitation events, for instance, have been shown to be related to a variety of weather systems, including extratropical cyclones (e.g., Kahana et al., 2002; Ulbrich et al., 2003; Kunkel et al., 2012; Grams et al., 2014), associated fronts (Catto and Pfahl, 2013) and warm conveyor belts (Browning, 1990; Pfahl et al., 2014), atmospheric rivers (Ralph et al., 2006; Lavers and Villarini, 2013), tropical moisture exports (Knippertz and Wernli, 2010; Knippertz et al., 2013), stratospheric potential vorticity streamers (e.g., Massacand et al., 1998; Martius et al., 2006) and cutoffs (e.g., Binder et al., 2021; Portmann et al., 2021; Röthlisberger et al., 2022), and mesoscale processes like deep convection (Mohr et al., 2020; Rüdisühli et al., 2020) and orographic ascent (Rotunno and Houze, 2007; Colle et al., 2013). Often, extreme precipitation also results from a complex interplay between some of these weather systems. Among them, extratropical cyclones have been shown to be particularly important for generating extreme weather conditions (e.g., Hawcroft et al., 2012; Pfahl and Wernli, 2012; Rüdisühli et al., 2020). In a climatological study based on reanalysis data, Pfahl and Wernli (2012) showed that a large fraction of 6-hourly extreme precipitation events coincides with extratropical cyclones in many regions of the world. Particularly high fractions were found in the storm track regions and in densely populated areas over the northeastern United States (US), the UK, northern Europe, around the Mediterranean Sea, and Japan. In addition, they showed that in the exit regions of the NH storm tracks and in most of the SH storm tracks cyclones causing extreme precipitation events are slightly more intense than those that do not cause extreme precipitation. Over the ocean and flat terrain, extreme precipitation often occurs near the cyclone centre, where dynamical lifting is strong, while extreme precipitation close to topography is typically related to cyclones at more remote locations that direct the moist flow toward the mountains (Pfahl, 2014). Several studies also showed that multi-day precipitation extremes can be linked to a serial clustering of extratropical cyclones, in particular at the downstream end of the NH storm tracks (e.g., Moore et al., 2020; Röthlisberger et al., 2022).

Often, extratropical cyclones are also accompanied by extreme near-surface winds. Well-known examples are the Queen Elizabeth II storm in 1978 off the east coast of the US (Gyakum, 1983), the Great October storm in the UK in 1987 (Hoskins



and Berrisford, 1988) and the windstorms Anatol, Lothar and Martin in December 1999 (Ulbrich et al., 2001; Wernli et al.,
2002) in Central Europe, whose intense winds all caused severe damage. High wind speeds can occur at different locations
within extratropical cyclones (Parton et al., 2010; Dacre et al., 2012; Hewson and Neu, 2015; Earl et al., 2017; Raveh-Rubin,
2017; Eisenstein et al., 2023; Gentile and Gray, 2023): close to the cyclone centre in the cold conveyor belt, in the warm sector
in the region of the warm conveyor belt, along the cold front co-located with convective precipitation, in the cold sector in
the descending dry intrusion, and, in some cyclones, in so-called sting jets, which are mesoscale descending airstreams at the
tip of the cloud head of the bent-back front (Browning, 2004). In addition, strong near-surface winds can occur relatively far
away from the cyclone centres in regions of pronounced surface pressure gradients, like for instance between cyclones and
blocking anticyclones (e.g., Pfahl, 2014), and over steep terrain related to orographic flow phenomena (Zardi and Whiteman,
2012; Jackson et al., 2013).

So far, only few studies investigated the nature and meteorology of extremes on the seasonal time scale, most of them
focusing on single case studies (e.g., Namias, 1978; Fischer et al., 2007; Cattiaux et al., 2010; Dole et al., 2014; Davies, 2015).
They highlighted the importance of both transient weather systems and longer-term climate variability for the occurrence
of anomalous seasons. Chang et al. (2015) analysed California winter precipitation over 36 past winters and found strong
correlations with extratropical cyclone activity over the Eastern North Pacific. Recent extremely dry Californian winters were
related to particularly low cyclone activity. In one of the first global climatological analyses, Flaounas et al. (2021) investigated
extremely wet seasons in a 40-year period in ERA-Interim reanalyses and showed that they often occur due to both an increased
number of wet days and of daily precipitation extremes, with the number of wet days being particularly important in arid regions
and daily extremes in regions of frequent precipitation like the tropics. They also showed that cyclones and warm conveyor
belts are anomalously frequent during extremely wet seasons in many parts of the world, but also an increased frequency of
tropical moisture exports and breaking Rossby waves can contribute to seasonal precipitation extremes in some areas. Both the
frequency and the precipitation efficiency of different weather systems are important for creating extreme seasonal precipitation
(Zschenderlein and Wernli, 2022). Boettcher et al. (2023) applied a recently developed approach for detecting extreme seasons
(Röthlisberger et al., 2021) to 70 years of ERA5 reanalyses to globally identify extremely windy and calm, hot and cold, and
wet and dry seasons as spatially coherent regions with extreme seasonal mean values of near-surface wind, temperature or
precipitation. They investigated the top 50 extreme seasons of each type and for each of the four seasons in the extratropics,
and showed that extremely dry seasons are on average associated with relatively few and weak cyclones, while extremely
wet seasons are associated with many, but not particularly intense cyclones. Extremely calm seasons exhibit slightly negative
cyclone frequency anomalies, but the cyclones are rather intense, while windy seasons are associated with positive anomalies
in terms of frequency and especially intensity. However, the patterns become less robust when only considering the top 10
extreme seasons of each type, suggesting substantial case-to-case and regional variability. This indicates that further research
is needed to understand the importance of cyclones and their characteristics for the occurrence of extreme seasons in different
regions of the world. In particular, for statistically robust statements about the meteorology of extreme seasons everywhere in
the extratropics, much longer datasets than those available from reanalyses are needed to yield a sufficiently large number of
extreme seasonal values at any specific location.



In this study, we aim to systematically quantify the occurrence frequency and characteristics of extratropical cyclones during
extremely wet, dry, windy and calm seasons in the NH and SH extratropics in winter and summer. To this end, we apply a
slightly adapted version of the approach by Röthlisberger et al. (2021) and Boettcher et al. (2023) to 1050 years of global
climate model data from the Community Earth System Model Large Ensemble (CESM-LE; Kay et al., 2015) to identify a large
number of extreme season objects everywhere in the extratropics in the present-day climate. Based on a systematic analysis of
extreme seasons in these 1050 years of climate simulations and two case studies of past extreme seasons in reanalysis data, we
address the following research questions:

1. Are extremely wet, dry, windy and calm seasons in the extratropics associated with anomalies in the frequency, intensity,
   origin, lifetime and stationarity of extratropical cyclones?

2. Are there any regional and seasonal differences in the cyclone characteristics during the different types of extreme
   seasons?

Several recent studies have shown that CESM is able to capture the geographical distribution, frequency, structure and
lifetime of extratropical cyclones and the associated precipitation-producing airstreams reasonably well (Raible et al., 2018;
Dolores-Tesillos et al., 2022; Binder et al., 2023; Joos et al., 2023), which indicates that the model is well suited for the purpose
of the present study. The remainder of the paper is structured as follows. Section 2 describes the data used in the study, the
method to identify extreme seasons, and the approach to attribute cyclone characteristics to extreme seasons. In Section 3,
we investigate the cyclone characteristics during two exemplary past extreme seasons over the UK in reanalysis data, the wet
winter in 2013/14 and the windy winter in 1988/89. Section 4 presents the results of the systematic analysis based on the
climate simulations, showing the cyclone characteristics during wet, dry, windy and calm winters and summers, respectively,
for each location in the extratropics. The results are summarised and discussed in Section 5.

## 2 Data and methods

### 2.1 Identification of extreme seasons in climate simulations and reanalysis data

We use present-day climate simulations from an initial condition ensemble of the Community Earth System Model (CESM),
version 1.2 (Hurrell et al., 2013), which was created with restart files from the CESM Large Ensemble (CESM-LE; Kay et al.,
2015), as described in Röthlisberger et al. (2020). Atmospheric fields are available every 6 h at a spatial resolution of $1.25°$
longitude by $\sim 0.9°$ latitude on 30 vertical levels. The simulations consist of three ensembles, each with 35 members. The first
35 members, referred to as the "macro ensemble", are direct reruns of the original 35 CESM-LE members, which were started
on 1 January 1920 and resimulated for the period from 1 January 1990 to 31 December 1999 to obtain high-resolution three-
dimensional model-level output. The second and third sets of 35 members, the "micro ensembles", were constructed by adding
tiny perturbations in the order of $10^{-13}$ K to the temperature field of the original CESM-LE members 1 and 2, respectively, on 1
January 1980. The model was then integrated forward for twenty years, but the first ten years of the simulations are ignored, to



**Table 1.** Total number of wet, dry, windy and calm extreme season objects and their mean area (km$^2$) identified in CESM in DJF (first number) and JJA (second number).

|  | Number of objects | Mean area (km$^2$) |
|---|---|---|
| Wet seasons | 14'444 / 15'068 | 355'984 / 316'790 |
| Dry seasons | 13'438 / 13'502 | 390'581 / 377'514 |
| Windy seasons | 9'644 / 11'300 | 566'030 / 468'217 |
| Calm seasons | 10'746 / 12'118 | 496'716 / 428'092 |

ensure that the spread in the atmospheric variables of the micro ensemble members is similar to the one of the macro ensemble members (see also Fischer et al., 2013). Thus, in total 105 ensemble members cover the 10-year time period from 1990 to 1999, which yields 1050 years of CESM data for the present-day climate that allow for the identification and statistical analysis of numerous extreme seasons.

To identify extreme (i.e, locally rare) seasons globally, we adopt a slightly modified version of the method developed by
Röthlisberger et al. (2021) and Boettcher et al. (2023). In a first step, we calculate at each grid point the seasonal mean values of precipitation and of the 10-meter wind speed for all 1050 years, separately for December to February (DJF) and June to August (JJA). In a second step, the 25 seasons with the lowest and the 25 seasons with the highest seasonal mean values are selected at each grid point as extremely dry and wet, calm and windy seasons, respectively. This corresponds to a local return period of 42 years. Finally, spatially coherent extreme season objects are formed by connecting neighbouring grid points
where the extreme values occur in the same season. To focus on significant events in the extratropics, we retain only objects with an area larger than $10^5$ km$^2$ whose centre of mass is located poleward of 30° latitude. At many oceanic grid points in the extratropics, the number of extreme season objects that overlap with the grid point remains close to 25, while over some mountainous areas the objects are smaller and the size filter reduces their number to about 10-15 events. The number of extreme season objects identified globally for each type and season and the mean size of the objects are shown in Table 1. Dry
and especially wet seasons are associated with many, typically rather small objects, while wind extremes are associated with fewer, but larger objects.

While the main focus of the paper lies on the statistical analysis of extreme seasons in climate simulations, in addition, two real examples of extreme seasons over the UK, a wet and a windy winter, respectively, are discussed. The seasons have been identified by Boettcher et al. (2023) in ERA5 reanalysis data from the European Centre for Medium-Range Weather
Forecasts (Hersbach et al., 2020), based on hourly data interpolated to a horizontal resolution of 0.5° for the years 1950-2020. The approach to identify the extreme seasons is similar to the one described above for the CESM data, with two exceptions: (i) calm and windy seasons are identified based on seasonal mean values of 10-m wind gusts rather than 10-m wind speed, and (ii) instead of choosing the events with the lowest and highest seasonal mean values, at each grid point the distribution over the 71 seasonal mean values is estimated based on statistical modelling, and the local return period is calculated. Seasons
with anomalies from the climatological mean that have a local return period of at least 40 years are considered to be extreme.





Spatially coherent objects are then formed by connecting grid points where the local return period exceeds 40 years in the same season (for details see Boettcher et al., 2023).

## 2.2 Cyclone characteristics during extreme seasons

Extratropical cyclones are identified and tracked using a slightly updated version of the method developed by Wernli and Schwierz (2006), which is described in Sprenger et al. (2017). Surface cyclones are identified as two-dimensional features outlined by the outermost closed isobar containing either a single or multiple local minima of sea level pressure (SLP). The maximum length of the outermost contour is restricted to 7500 km. The tracking algorithm searches for each SLP minimum the most likely continuation among the minima identified at the next time step within a given search area. Only cyclones with a minimum lifetime of 24 h are included in the analysis.

For each extreme season object, cyclone frequencies and characteristics are computed during the extreme season and in the climatology. To this end, the cyclones are identified whose centres (defined as their SLP minimum) moved over any part of the object area during the extreme season. For each extreme season object, the same procedure is also done for all 1050 seasons in CESM and, for the ERA5 case studies, for all 71 seasons between 1950-2020 in order to obtain the climatology of cyclone characteristics for each extreme season object. The number of cyclones is used as a measure for cyclone frequency. For each cyclone crossing an extreme season object, several additional characteristics like intensity, intensification rate, origin, lifetime and stationarity are calculated. Intensity is measured by the minimum SLP along the track, and the intensification rate as the latitude-adjusted maximum change in SLP within 24 h along the track in units of Bergeron (Sanders and Gyakum, 1980). The minimum SLP and the maximum deepening rate are determined along the entire cyclone track, but the results are very similar if only the segment of the track is considered that moved over the extreme season object. Cyclones with a deepening rate of more than 1 Bergeron (which corresponds to 24 hPa in 24 h at $60°$ latitude) are considered to intensify explosively as so-called "bombs", and their frequency is also examined during extreme seasons and the climatology. Furthermore, we investigate the genesis latitude and longitude of the cyclones, as well as the total lifetime and the time the track spent within the extreme season object, which we use as a measure for stationarity.

By including all cyclones in the analysis whose centres moved over any part of the extreme season object rather than just those that moved over a specific grid point, we take into account remote effects of cyclones and the fact that the entire cyclone area as well as elongated fronts far away from the cyclone centre can be associated with strong precipitation and wind gusts. Nevertheless, not all remote effects are included, as the approach ignores those cyclones where part of the area or a front overlapped with the object but not the cyclone centre, even if they may also have contributed to extreme weather in the object area. Although the method certainly ignores the contribution of important cyclones, it avoids the need to define an arbitrary and subjective threshold to decide how much overlap between cyclone area and object is necessary to include a cyclone in the analysis. And as we apply the same approach during extreme seasons and the climatology, the resulting differences provide meaningful information about cyclone anomalies during extreme seasons.





## 2.3 Anomaly maps of cyclone characteristics and statistical significance testing

In CESM, global maps of the differences in the various cyclone characteristics between extreme seasons and the climatology
are produced by computing for each grid point the average characteristics over all $N$ extreme season objects that include the
grid point (e.g., the average number of cyclones or the average minimum SLP of the cyclones), separately for the extreme
seasons and for the climatology. To detect statistically significant differences, a bootstrapping procedure is performed at each
grid point and for each cyclone characteristic. Hereby, we assume that the specific characteristic, like, for instance, the number
of cyclones, does not differ between extreme seasons and the climatology, i.e., that it belongs to a common distribution. Under
this null hypothesis, we randomly select 1000 times $N$ seasons among the total number of 1050 seasons and each time calculate
their mean cyclone characteristics. The mean value over the $N$ extreme seasons is then ranked among the 1000 bootstrapping
values, and the two-sided local $p$ value is calculated. As the hypothesis test is applied to each grid point, to account for spatial
correlation the $p$ values are adjusted by controlling the false discovery rate at $\alpha_{FDR} = 0.1$, following Benjamini and Hochberg
(1995) and Wilks (2016). All $p$ values smaller than $\alpha_{FDR}$ are considered to be statistically significant and stippled in the
figures in Section 4.

## 3 Two exemplary extreme seasons in ERA5

### 3.1 The wet winter 2013/14 over the UK and Ireland

As mentioned in the introduction, during winter 2013/14 Ireland, large parts of the UK and the eastern North Atlantic expe-
rienced unusually strong precipitation that resulted in widespread ecological and economic damages (e.g., Kendon and Mc-
Carthy, 2015). The red contour in Fig. 1a outlines the region identified as an extremely wet winter in ERA5. Also shown are
the tracks of the 58 cyclones whose centres crossed the extreme season object during winter 2013/14. The cyclones typically
originated in the western North Atlantic and then moved northeastward along the storm track into the eastern North Atlantic,
where they attained their minimum SLP and eventually dissolved. Substantially more cyclones went through the area of the
extreme season object in winter 2013/14 than during most other seasons in the 1950-2020 climatology, with the number in
2013/14 corresponding to approximately the $95^{th}$ percentile of the historical data (Fig. 1b). Even more remarkable is the high
frequency of explosively deepening cyclones (i.e., bomb cyclones) that occurred during the extreme winter, which exceeded the
mean climatological frequency by more than a factor of two (Fig. 1c). Accordingly, the minimum SLP of most cyclones was
also lower and their deepening rate was higher (Fig. 1d,e). In addition, in the mean, the cyclones during the extreme season had
a further southward and westward origin (Fig. 1f,g), a longer lifetime (Fig. 1h), and they spent a larger number of time steps
within the area of the extreme season object, i.e., they were more stationary than usual (Fig. 1i). The results are consistent with
Matthews et al. (2014), who found that the combination of both high cyclone frequency and high cyclone intensity resulted in
the stormiest winter in a 66-year record over Ireland and the UK. As shown by Davies (2015), the high cyclone frequency was
linked to an anomalously strong and persistent jet stream.



**Figure 1.** (a) Extremely wet winter object in 2013/14 (red contour) and associated cyclone tracks (grey lines). The blue and yellow dots mark the cyclogenesis and lysis positions, respectively, and the green dots mark the positions where the cyclones attained their minimum SLP. (b) Number of cyclones per season and (c) number of bombs per season whose centres crossed the object (red contour in a) in winter 2013/14 and in the climatology (all winter in the period 1950-2020). (d) Minimum SLP (hPa) of the cyclones, (e) deepening rate (Bergeron), (f) genesis latitude (°N), (g) genesis longitude (°W), total duration of the cyclone tracks (hours), and (h) stationarity (hours). The boxes extend from the $25^{th}$ to the $75^{th}$ percentile and the whiskers from the $5^{th}$ to the $95^{th}$ percentile of the data. The line represents the median and the black dot the mean.





**Figure 2.** As Fig. 1, but for the windy winter 1988/89.

## 3.2 The windy winter 1988/89 over the UK and Scandinavia

The region over the eastern North Atlantic, the northern part of the UK and large parts of Scandinavia, outlined in Fig. 2a, was
identified in ERA5 as an extremely windy winter in 1988/89. Particularly noteworthy for this winter is an intense cyclone that
moved from the north of the UK toward Norway on 13 February 1989, causing widespread damage and record-strong wind
gusts in Scotland (Burt, 2021; Met Office, 2024). The cyclone statistics over the entire season show that, in contrast to the wet
winter discussed above, the number of cyclones in this windy winter was reduced compared to the climatology (Fig. 2b), and
on average they were slightly less stationary in the region of the extreme season object (Fig. 2i). (Because this extreme season





object is larger than the one portrayed in Fig. 1, the climatological cyclone frequency is higher in Fig. 2b compared to Fig. 1b.) On the other hand, as for the wet winter the cyclones were substantially more intense than usually, with a higher number of bombs (Fig. 2c), a deeper minimum SLP (Fig. 2d) and a stronger intensification rate in the mean (Fig. 2e), and they typically originated from further south and west (Fig. 2f,g), and had longer tracks than in the climatology (Fig. 2h). Thus, while for

the wet winter in 2013/14 both the increased cyclone frequency and the increased intensity were important, the windy winter in 1988/89 was associated with relatively few, but particularly intense cyclones, like the one that hit Scotland on 13 February 1989.

## 4    Statistical analysis of extreme seasons in CESM

### 4.1    Extremely wet seasons

After looking at two exemplary ERA5-based extreme seasons that affected the UK, we investigate the cyclone characteristics of a large number of extreme seasons around the globe in 1050 years of present-day climate in the CESM data set. In this Section, extremely wet seasons are discussed. Figure 3 shows the anomaly in cyclone frequency and intensity in extremely wet seasons with respect to the climatology in DJF. As described in Sect. 2.3, at each grid point the average cyclone characteristics are computed over all $N$ extreme season objects that contain the grid point, separately for the extreme seasons and the climatology.

For instance, the grid point at 53°N, 7.5°W over Ireland overlaps with $N = 17$ wet extreme season objects during DJF. For each of these objects, the cyclones are counted that crossed the object and their characteristics are calculated in the same way as for the two case studies discussed in the previous section. Averaged over all 17 objects, the mean number of cyclones that moved over the respective object is 16.2 during the extremely wet seasons and 12.3 in the climatology (i.e., averaged over all 1050 seasons), and they were associated with a mean minimum SLP of 970 hPa and 977 hPa, respectively. The differences

between wet seasons and the climatology, i.e., +3.9 cyclones and −7 hPa, are shown in Figs. 3a and 3c at 53°N, 7.5°W. Overlaid in Fig. 3a is also the mean climatological cyclone frequency (i.e., 12.3 cyclones at the grid point over Ireland). As mentioned in Sect. 2.2, it is important to note that the centres of these cyclones did not necessarily move over this grid point – they could have moved over any part of the extreme season objects. This takes into account that the entire cyclone area and associated fronts can contribute to precipitation at a specific grid point. The cyclone numbers during the extreme seasons and

in the climatology depend on the area of the extreme season objects, with larger objects typically yielding larger numbers (see also examples in Sect. 3). The climatological cyclone frequency therefore differs between different types of extreme seasons (i.e., wet, dry, windy, and calm seasons) and between DJF and JJA.

According to Fig. 3a, the anomalies in cyclone number during extremely wet seasons are positive in large parts of the extratropics, and the stippling indicates that the differences are statistically significant almost everywhere. Particularly large

anomalies can be found over most ocean basins, especially in regions with high climatological cyclone frequencies (blue contour in Fig. 3a), and over southeastern Europe and parts of Russia. Exceptions with no or only weak anomalies occur over some continental regions like the central US, China and Antarctica, which are all regions with relatively low climatological



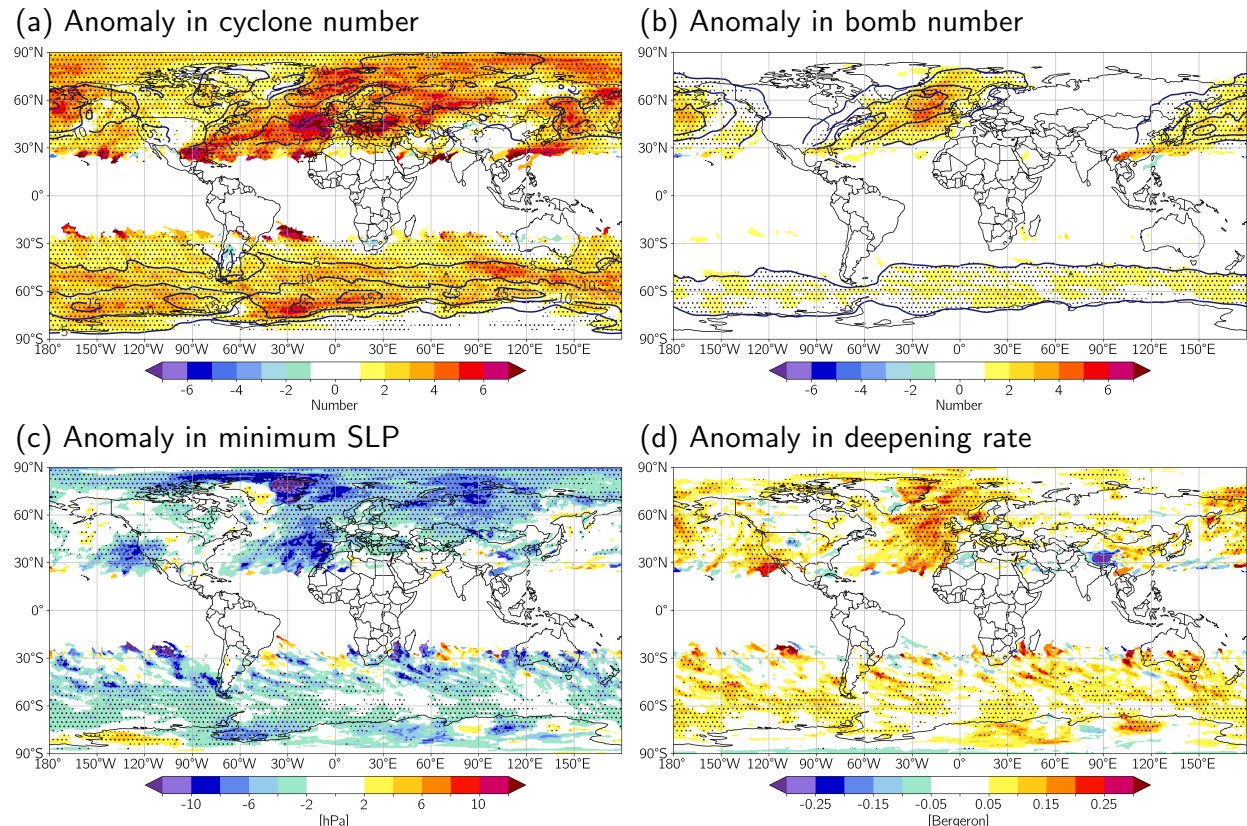

**Figure 3.** Anomalies in cyclone characteristics during extremely wet DJF seasons with respect to the climatology in CESM-LE. The shading shows the seasonal anomalies in the (a) number of cyclones, (b) number of bomb cyclones, (c) mean minimum SLP of the cyclones (hPa), and (d) mean deepening rate of the cyclones (Bergeron). (See text for details about how the anomalies are computed). In addition, the blue contours in (a, b) show the values for the climatology, i.e., the mean climatological number of (a) cyclones and (b) bombs per season averaged over the extreme season objects that contain the grid point. The values in (b) are only shown for regions where the mean climatological bomb frequency is larger than 1 bomb per season. Stippling denotes areas where the differences between extreme seasons and the climatology are considered to be statistically significant.





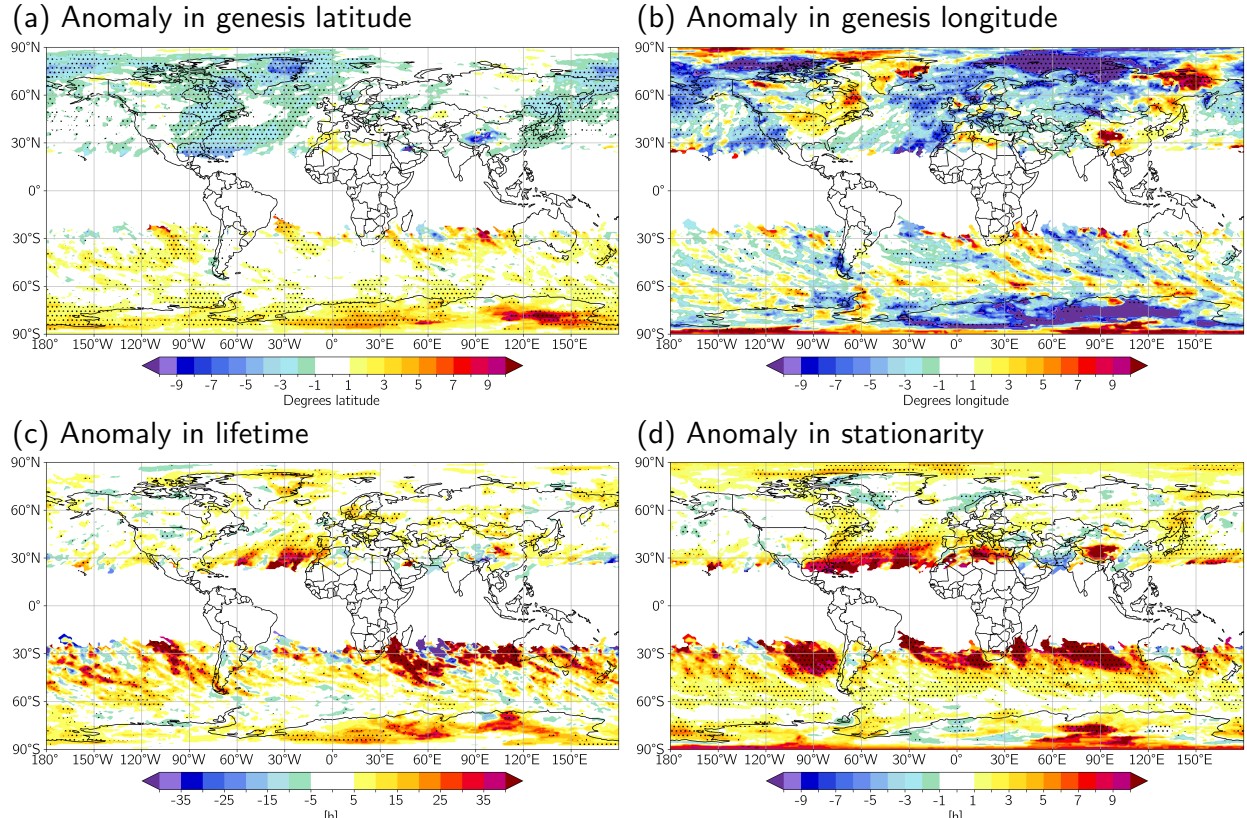

**Figure 4.** As Fig. 3, but showing, again for extremely wet DJF seasons, anomalies in the cyclones' (a) genesis latitude (degrees), (b) genesis longitude (degrees), lifetime (hours), and (d) stationarity, measured by the number of time steps within the extreme season object (hours).

cyclone frequencies, as well as parts of Greenland and southern Australia. Near the cyclone frequency maximum east of the Andes over Argentina, there is even a small region with negative anomalies in cyclone number.

Over the North Atlantic and North Pacific and in many coastal regions at the beginning and end of the storm tracks, extremely wet seasons in DJF are not only associated with anomalously many cyclones in general, but also with anomalously many explosively deepening cyclones (Fig. 3b). The same is true for the SH storm track, but there, the anomalies are smaller than in the NH, as also the absolute numbers are smaller in the summer hemisphere. Over the majority of the continents, bomb cyclones occur very rarely and the anomalies are not shown.

Concomitant with the large bomb cyclone frequency, over the eastern parts of the NH ocean basins and the adjacent land areas, as well as over large parts of the Southern Ocean the mean minimum SLP of the cyclones is lower (Fig. 3c) and the deepening rates are higher (Fig. 3d) during wet seasons than in the climatology. Anomalously deep cyclones during wet seasons also occur over the Arctic ocean, eastern Greenland, parts of the Mediterranean, Southeastern Europe and the majority of Russia (Fig. 3c). Thus, in these regions extremely wet seasons are the result of a combination of anomalously many and anomalously





intense cyclones, consistent with the findings for the wet winter 2013/14 over the UK and Ireland (Sect. 3.1). In contrast, over the western North Atlantic and North Pacific, most of North America, central Europe, eastern Russia and, in the SH, in the latitudinal band between about 30°S and 50°S, cyclones are not particularly intense during extremely wet seasons. Here, wet DJF seasons are the result of anomalously many, but not especially deep cyclones.

Cyclones contributing to wet DJF seasons often originate a few degrees further equatorward than usually (Fig. 4a), in particular for extreme seasons in the western and central part of the NH ocean basins, and they originate a few degrees further west for extreme seasons in the central and eastern part of the basins (Fig. 4b). In terms of cyclone lifetime, in most regions of the world there is no statistically significant signal, except for the southeastern North Atlantic with positive anomalies (Fig. 4c). Cyclones are anomalously stationary in wet seasons along the US east coast, over the subtropical North Atlantic, in the eastern Mediterranean, the western North Pacific and in parts of the Southern Ocean. Except the US east coast, these are all regions with high climatological frequencies of potential vorticity cutoffs (Portmann et al., 2021). Cutoffs are often accompanied by surface cyclones and precipitation. They can be relatively long-lived and stationary and thereby result in the formation of long-lasting heavy precipitation events (e.g., Doswell et al., 1998; Grams et al., 2014; Röthlisberger et al., 2022).

As mentioned above, some continental regions like the central US, China, western Greenland and large parts of Argentina, southern Australia and Antarctica are not associated with anomalously high cyclone frequencies during extremely wet seasons (Fig. 3a). Also, they are not associated with particularly intense or stationary cyclones (Fig. 3c, Fig. 4d). The regions are all located close to topography, where precipitation is typically orographically induced. In addition, cyclones at more remote locations outside the extreme season objects could contribute to heavy precipitation in these regions. As shown by Pfahl (2014), extreme precipitation over complex terrain in Europe is typically associated with cyclones that are located relatively far away from the location of the precipitation, whereby the cyclones are important for directing moist air toward the topography. Furthermore, the extreme precipitation might also be related to other weather systems like atmospheric rivers and tropical moisture exports. Over Argentina and southern Australia, summertime deep convection most likely plays an important role for extreme seasonal precipitation during DJF, explaining the absent signals in cyclone characteristics.

In JJA, extremely wet seasons in the SH exhibit relatively similar patterns to those in DJF, with slightly stronger anomalies in most regions in terms of cyclone number and intensity (Fig. 5a,b vs. 3a,c), and in particular in terms of the number of bomb cyclones (not shown). In the NH, the cyclone frequency anomalies during wet summers are also positive over most oceanic regions and over eastern Canada, northern Europe, most of Russia, eastern China, and Japan, but weaker than in DJF. Negative cyclone frequency anomalies during summertime extreme seasons occur in some low-latitude orographic regions like the central US, the Iberian Peninsula and the Tibetan Plateau, and no anomalies are found over most other land areas and the Mediterranean ocean. As further discussed in Sect. 4.2, the regions with negative anomalies in Fig. 5a are climatologically characterised by relatively frequent heat lows, which can be strong enough to be identified by our algorithm. Heat lows are typically dry, which could explain their reduced frequency during extremely wet seasons. In contrast to DJF, there are no anomalies in cyclone intensity in most of the NH during wet summers except in subpolar and polar regions like Alaska, Greenland, Scandinavia and the Arctic ocean. The absence of positive cyclone frequency and intensity anomalies during wet summers over most subtropical and mid-latitude continental regions indicates the importance of other mechanisms for trigger-





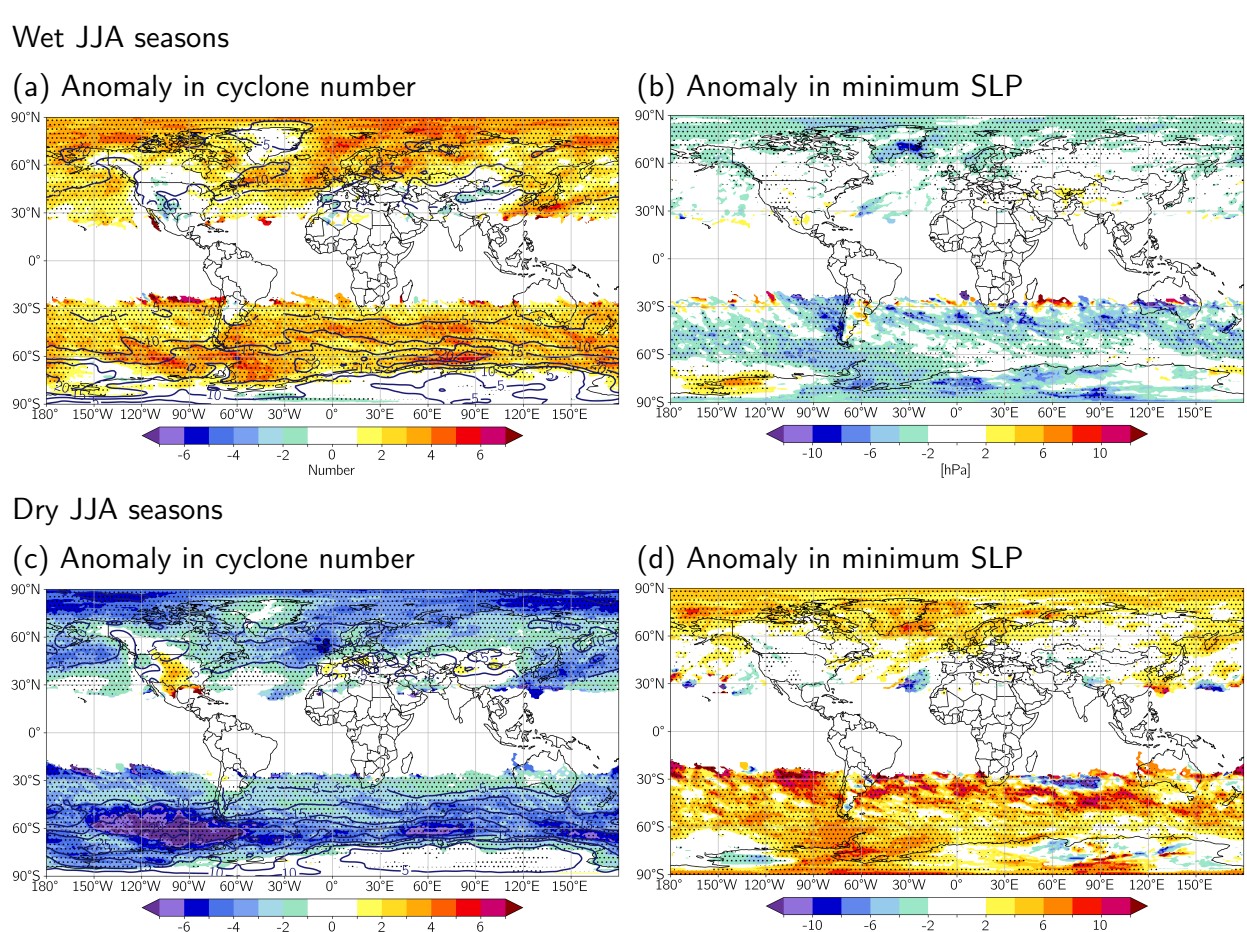

**Figure 5.** Anomalies in cyclone characteristics during extremely (a, b) wet JJA and (c, d) dry JJA seasons. Fields as in (a, c) Fig. 3a and (b, d) Fig. 3c.



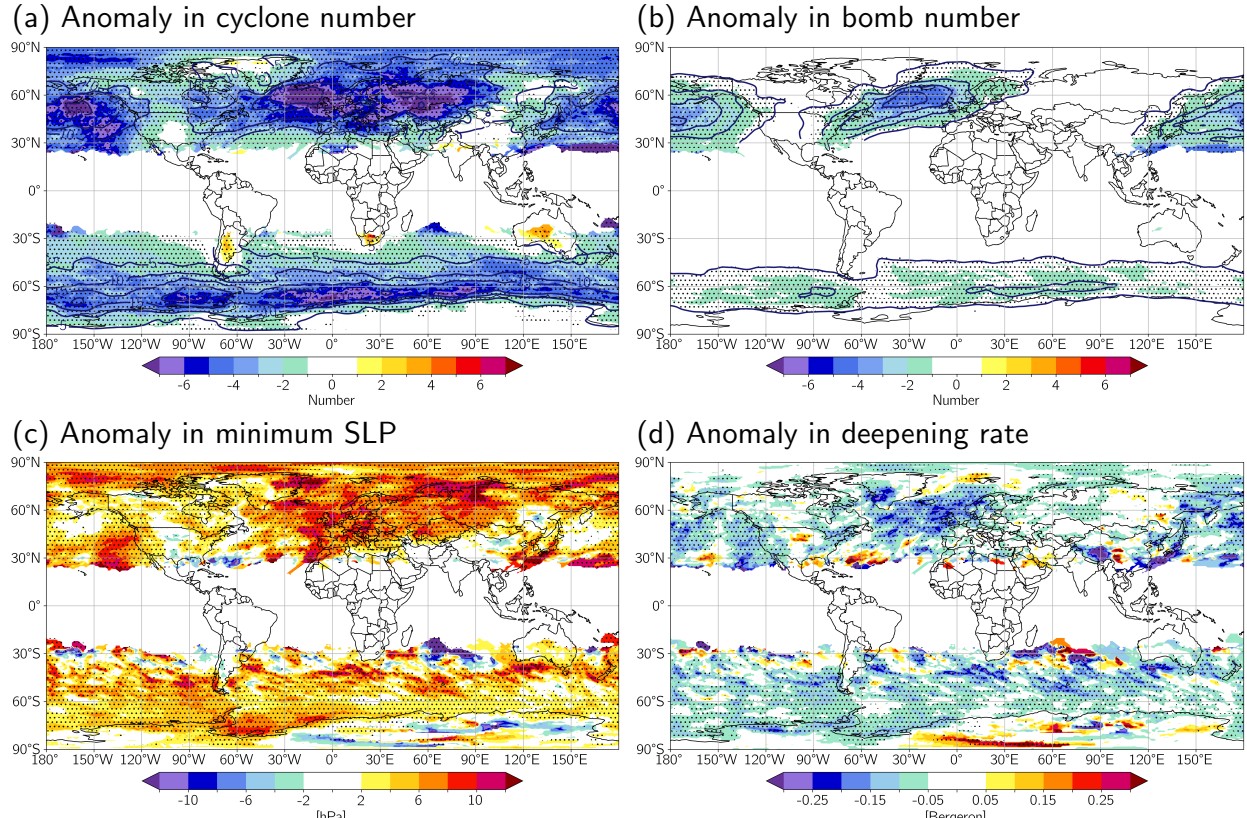

**Figure 6.** As Fig. 3, but for extremely dry DJF seasons.

ing extreme precipitation like summertime deep convection, orographic lifting and, over South and East Asia, anomalies in the Asian monsoon. In addition, anomalously stationary cyclones contribute to extremely wet summers over the exit regions of the NH storm tracks and large parts of central, eastern and southeastern Europe, the eastern Mediterranean and Kazakhstan (not shown).

## 4.2 Extremely dry seasons

Extremely dry seasons mainly exhibit the opposite patterns to extremely wet seasons. In DJF, they are associated with anomalously low cyclone frequencies in most of the world and low frequencies of bomb cyclones in the storm track regions (Fig. 6a,b). Thus, in many areas extremely dry seasons result from the absence of cyclones. In large parts of the extratropics the cyclones that still occur in extremely dry seasons have anomalously weak intensities and intensification rates (Fig. 6c,d), and they often originate from further poleward and eastward (or from less far to the south and west) and have shorter lifetimes than in the climatology (not shown). In addition, cyclones contributing to dry seasons are less stationary than climatologically in many parts of the extratropics, for instance over the Mediterranean and the adjacent land areas, over the entrance and exit regions of





the NH storm tracks, over the Barents Sea and in parts of the Southern Ocean, while they are more stationary than usually over Alaska and Siberia (not shown).

In JJA, extremely dry seasons are also associated with anomalously low cyclone frequencies over most of the NH and SH
oceans as well as northern and eastern Canada, northern Europe, most of Russia, eastern China, and Japan, while the central US, the Mediterranean and the adjacent land areas, South and East Asia are associated with no or even positive cyclone frequency anomalies (Fig. 5c). These positive anomalies (see also similar signals in Fig. 6a in the SH) might be an indication of heat lows. As heat lows are typically dry, their increased frequency might be conducive to the formation of an extremely dry season. This hypothesis is supported by the fact that the cyclones in these regions are anomalously stationary during dry seasons (not
shown), a characteristic typical of heat lows. In the majority of the SH extratropics and in subpolar and polar regions in the NH, cyclones during dry JJA seasons have anomalously weak intensities, while no anomalies are found over most of the NH midlatitudes (Fig. 5d). As in DJF, cyclones are less stationary than usually in many parts of the extratropics, in particular over the exit regions of the NH storm tracks, and they often originate from anomalously far poleward and eastward and have anomalously short lifetimes (not shown).

**4.3  Extremely windy seasons**

The anomalies in cyclone frequency and intensity of extremely windy DJF seasons with respect to the climatology are shown in Fig. 7. The climatological cyclone and bomb cyclone frequencies associated with the windy objects (blue contours in Fig. 7a,b) are higher than those associated with the wet objects (Fig. 3a,b), as the area of the windy objects is typically larger (Table 1). Windy seasons are associated with anomalously deep cyclones almost in the entire extratropics (Fig. 7c). In the NH storm
tracks, they also coincide with anomalously many bomb cyclones (Fig. 7b), and, both in the NH and SH storm tracks, anomalously strong deepening rates (Fig. 7d). The cyclone frequency anomaly pattern of extremely windy seasons (Fig. 7a) looks different to the one of extremely wet seasons (Fig. 3a). In contrast to wet seasons, windy seasons in DJF are associated with anomalously few cyclones in many regions of the world, in particular in the main SH storm track between 40°S and 70°S, in large parts of the North Pacific, the northeastern North Atlantic, the northern part of the UK, Scandinavia, western Russia
and the Barents-Kara Sea (Fig. 7a). Hence, windy DJF seasons are often the result of relatively few, but particularly intense cyclones, consistent with the above discussed windy winter 1988/89 over the UK and Scandinavia (Sect. 3.2). In contrast, positive cyclone frequency anomalies during windy seasons occur in the southern part of the North Atlantic storm track in a band extending from the Gulf of Mexico to the Mediterranean, along the US west coast, over Canada, and to the north and the south of the main storm track in the SH. In these regions windy seasons result from the combination of unusually many
and unusually deep cyclones. The positive cyclone frequency anomalies on the southern edge of the North Atlantic storm track suggest that windy seasons in these regions occur during anomalous equatorward deviations of the storm track and most likely also the upper-level jet. Similarly, windy seasons around 30°S and 75°S are most likely linked to equatorward and poleward shifts, respectively, of the SH storm track. Some continental regions close to topography like the central US, Alaska, Greenland, China, eastern Russia and Antarctica are neither associated with more nor with deeper cyclones during extremely windy
seasons, a similar pattern to the one found for wet seasons (Fig. 3a,c). In these regions strong winds are probably significantly



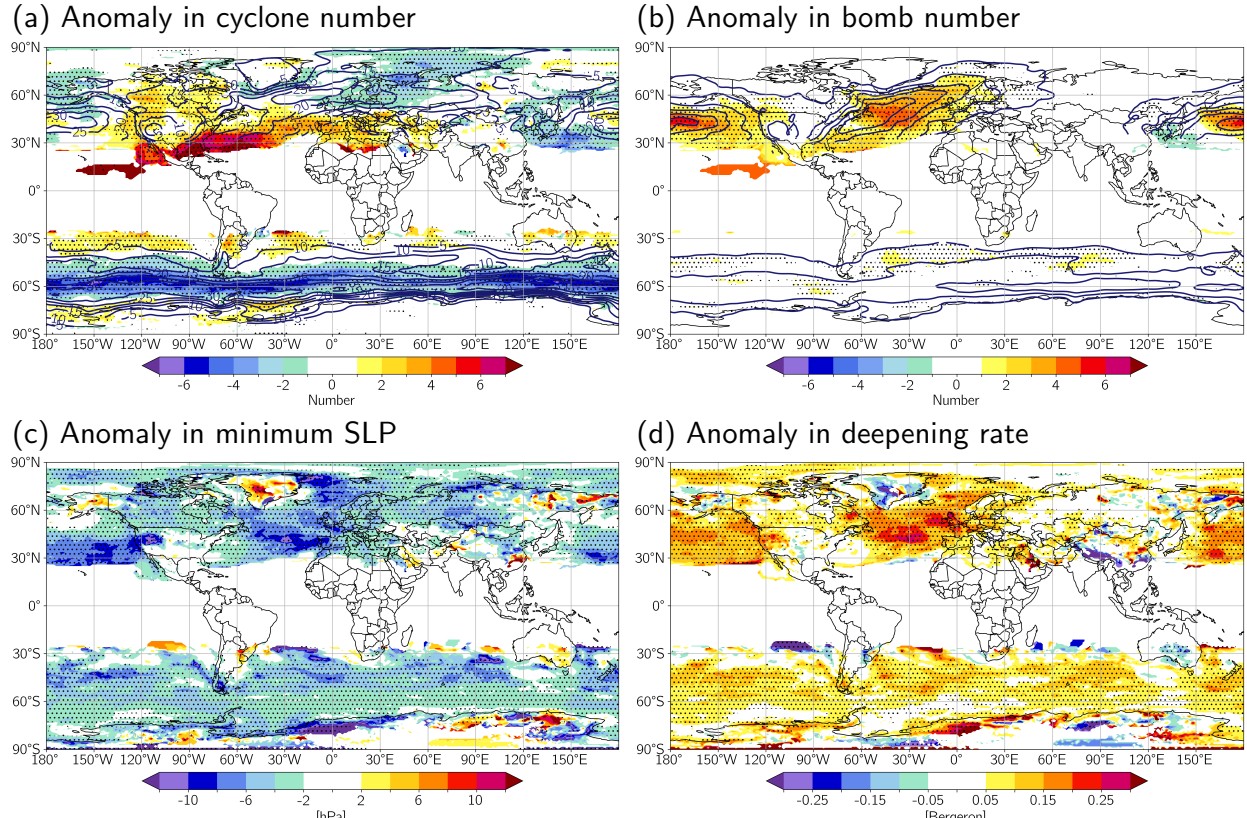

**Figure 7.** As Fig. 3, but for extremely windy DJF seasons.

influenced by mountain effects and orographic flow systems like, for instance, mountain waves, downslope windstorms and mountain-valley winds (Zardi and Whiteman, 2012; Jackson et al., 2013). Although there are no anomalies in cyclone frequency and intensity in these regions, cyclones at remote locations outside of the extreme season objects could also contribute to strong winds in these regions by directing the flow toward the topography, as for heavy precipitation.

In many regions of the world (e.g., over the main ocean basins, the western US and western Europe), cyclones originate some degrees further to the west than usually during windy DJF seasons (Fig. 8b) and, in particular over the NH ocean basins, they live about 5-15 h longer (Fig. 8c). The anomaly pattern in genesis latitude is less coherent but indicates negative anomalies (i.e., a more equatorward origin) at the entrance of the North Atlantic storm track and over the UK (Fig. 8a). Cyclones contributing to windy seasons are more stationary than usually in the western North Atlantic, but less stationary in many other regions of the
world like the SH storm track, the central North Pacific, the northeastern North Atlantic, the Mediterranean, the Arctic ocean, and central Europe (Fig. 8d). The fast motion is consistent with the high intensification of the cyclones (Fig. 7d).

In JJA, windy seasons are associated with anomalously few, but intense cyclones in almost the entire SH midlatitudes (Fig. 9a,b), and they originate further west and move faster than in the climatology (not shown). In the NH, fewer, but more





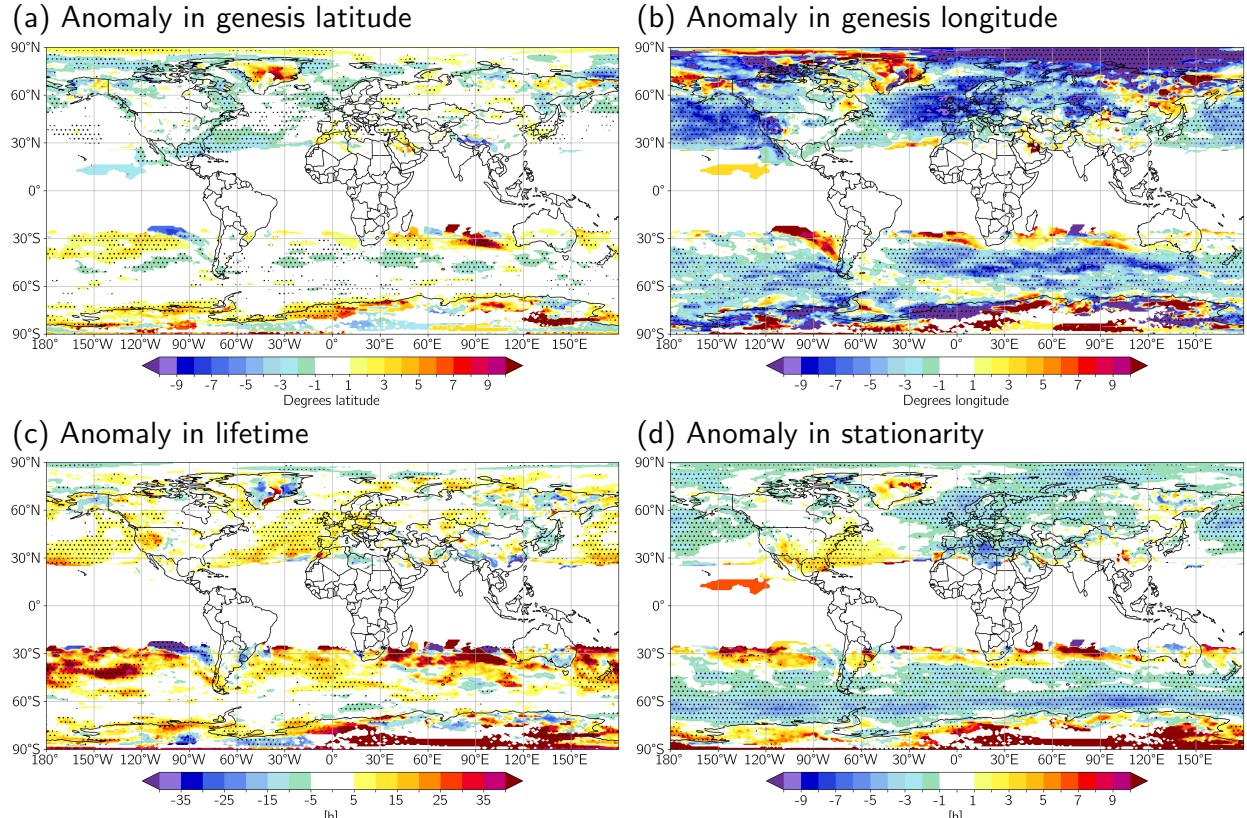

**Figure 8.** As Fig. 4, but for extremely windy DJF seasons.

intense cyclones also occur over parts of the North Pacific. The North Atlantic, the Arctic ocean and Canada are associated both

with more and deeper cyclones, while the central and eastern US and parts of Europe are associated with more, but not with deeper cyclones. Over most of Asia, in the western US, the Mediterranean and Greenland windy summers are neither associated with more nor with deeper cyclones. In these regions, the windy seasons probably result from anomalies in the frequency and intensity of other mechanisms that can cause strong winds, like heat lows, summertime convection and orographic effects, or they are related to cyclones at more remote locations.

**4.4   Extremely calm seasons**

Of the four different types of extreme seasons, calm seasons might appear as the least impactful. However, particularly for the energy sector, which increasingly relies on renewable sources, seasons with minimum near-surface winds pose growing challenges. In extremely calm seasons, many of the anomaly patterns contrast with those observed during extremely windy seasons. In DJF, in most of the extratropics the cyclones are less intense (Fig. 10c). Exceptions without any significant anomalies are

again some continental regions close to topography like the central US, Greenland, China, eastern Russia, and Antarctica. Cy-



Windy JJA seasons

(a) Anomaly in cyclone number

(b) Anomaly in minimum SLP

Calm JJA seasons

(c) Anomaly in cyclone number

(d) Anomaly in minimum SLP

**Figure 9.** As Fig. 5, but for extremely (a, b) windy and (c, d) calm JJA seasons.

clones typically have weaker intensification rates (Fig. 10d), and in the NH storm track regions the number of bomb cyclones is strongly reduced (Fig. 10b). The cyclone frequency anomalies are negative in large parts of the NH extratropics and on the northern and southern flanks of the SH storm track (Fig. 10a). Thus, in these regions calm seasons are typically the result of unusually few and weak cyclones. In contrast, positive cyclone frequency anomalies occur in the main SH storm track, i.e.,

more, but weaker cyclones occur during calm seasons there.

Calm JJA seasons are also associated with anomalously weak cyclones throughout the SH extratropics and in large parts of the NH ocean basins, Canada, and parts of Europe, but the anomalies in the NH are weaker than during DJF (Fig. 9d). The positive cyclone frequency anomalies in the SH storm track are weaker and extend over a smaller longitudinal area than in DJF, with negative anomalies in large parts of the SH extratropics (Fig. 9c). In the NH, negative cyclone frequency anomalies

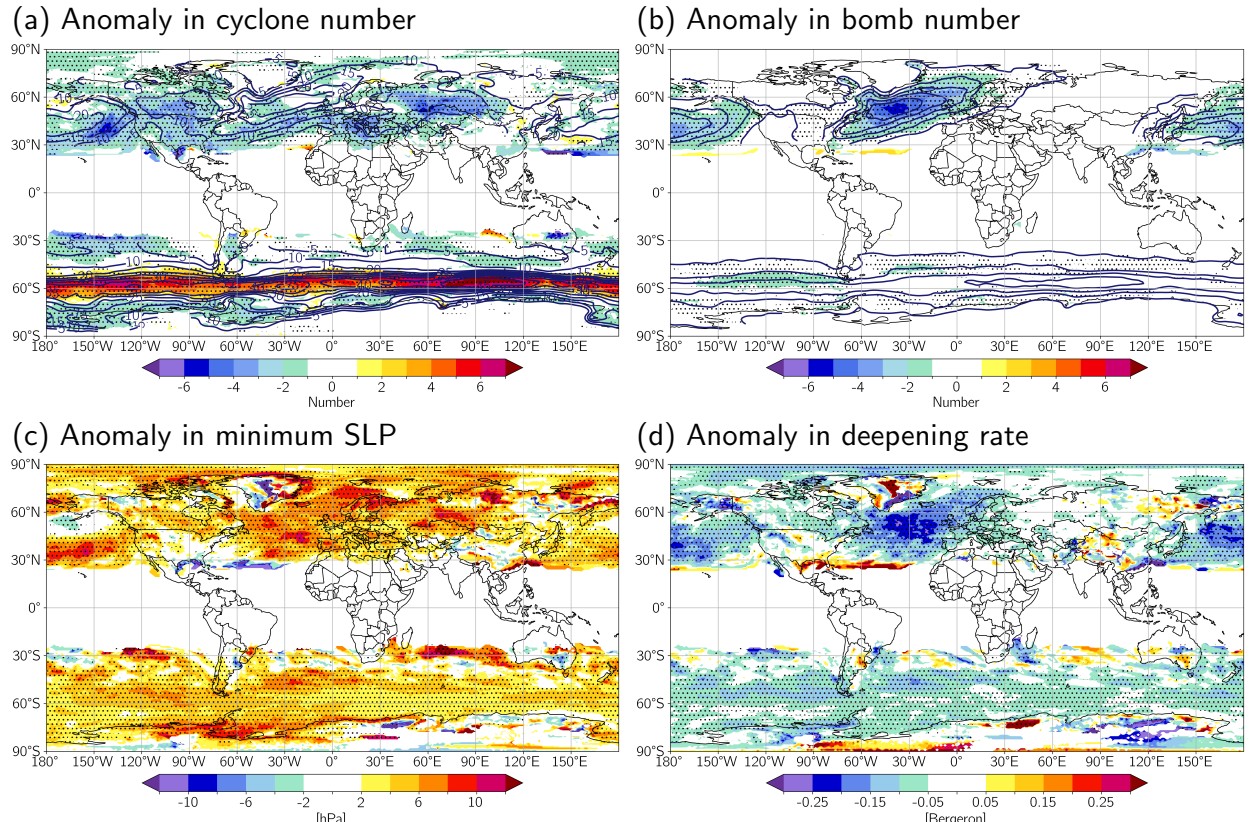

**Figure 10.** As Fig. 3, but for extremely calm DJF seasons.

are found in many oceanic regions and over central and eastern North America, eastern Europe, and eastern China. Similar to windy summers, in many continental regions (e.g., large parts of Asia, the US west coast, and Greenland) and over the Mediterranean, calm summers are neither associated with strong anomalies in cyclone frequency nor intensity, which further corroborates that the presence or absence of strong winds in these regions is, in summer, mainly related to other phenomena than extratropical cyclones. In both seasons, in many parts of the world and in particular over the oceans, cyclones contributing
to calm seasons originate further eastward, have shorter lifetimes, and are more stationary than in the climatology (not shown).

## 5 Summary and conclusions

In this study, we investigate the importance of extratropical cyclones and their characteristics for the occurrence of extremely wet, dry, windy and calm seasons in the extratropics in 1050 years of present-day (1990-2000) climate simulations of the Community Earth System Model Large Ensemble (CESM-LE) and in two exemplary case studies of past extreme seasons over
the UK in ERA5 reanalyses. Extreme seasons are defined as spatially coherent regions of extreme seasonal mean values of





precipitation or near-surface wind, with a return period of about 40 years (Röthlisberger et al., 2021; Boettcher et al., 2023). For each extreme season object, the cyclones are identified that crossed the object during the extreme season and during each other season in the climatology, and the characteristics of these cyclones like intensity, origin, lifetime and stationarity are compared. In CESM, each grid point overlaps with about 10-25 extreme season objects, which allows for a detailed statistical
analysis of the associated cyclone characteristics everywhere in the extratropics. Over the UK and the eastern North Atlantic, the results based on CESM qualitatively agree with those of the two illustrative extreme season case studies investigated in ERA5, the wet winter 2013/14 and the windy winter 1988/89. This indicates that the climate model captures the properties of extratropical cyclones reasonably well, which is in line with previous studies that investigated cyclones and their associated warm conveyor belts in CESM-LE simulations (Binder et al., 2023; Joos et al., 2023).

We now summarise and discuss the main findings for extremely wet and windy seasons. With a few exceptions mentioned in Sect. 4, extremely dry and calm seasons have reversed cyclone properties compared to wet and windy seasons, and therefore do not require a separate discussion. The next paragraphs first look at the main storm track regions (where cyclones are frequent), then at oceanic regions at the edge of the storm tracks, and finally at selected land regions.

In the main storm track regions, extremely wet winters (DJF for the NH and JJA for the SH) are associated with positive
anomalies in cyclone and bomb cyclone frequencies. At the downstream ends of the NH storm tracks and in parts of the SH storm track, the cyclones are also anomalously intense and have a further westward origin than usually. In contrast to extremely wet winters, windy winters are associated with negative cyclone frequency anomalies in most of the main storm track regions. However, the cyclones are significantly more intense than climatologically, with an anomalously high number of bomb cyclones, and they originate further to the west, move faster and have longer lifetimes. Thus, a few exceptionally strong
storms can result in extreme seasonal wind anomalies in the main storm track regions. In summer, extremely wet seasons are characterised by anomalously many cyclones and windy seasons by anomalously intense cyclones in most of the main storm track regions, but the anomalies are weaker than in winter. In oceanic regions at the equatorward edges of the storm tracks, anomalously many and stationary cyclones occur during extremely wet seasons both in winter and summer. Windy seasons in these regions are often characterised by anomalously many, deep and long-living cyclones, in particular at the southern edge of
the North Atlantic storm track during winter and on the southern and northern edges of the SH storm track during winter and summer. They are most likely linked to equatorward and poleward shifts, respectively, in the position of the storm track and the upper-level jet.

The cyclone anomaly patterns for selected land regions are summarised in Table 2 for local winter and Table 3 for local summer. They highlight strong differences between the regions, between winter and summer and between wet and windy sea-
sons. For instance, during extremely wet winters California is associated with medium-strong positive anomalies in cyclone frequencies and deepening rates and strong negative anomalies in minimum SLP (i.e., particularly deep cyclones), the central US is not associated with significant anomalies in any of the considered cyclone characteristic, while Florida is associated with strong positive anomalies in cyclone frequencies and stationarity and medium-strong positive anomalies in the frequency of bomb cyclones. Wet winters over the UK are characterised by anomalously many and deep cyclones, over Portugal they are
characterised by anomalously many, deep, strongly deepening and long-living cyclones, and over the Mediterranean by anoma-





**Table 2.** Cyclone characteristic anomalies of extreme seasons in specific regions in local winter (DJF, except JJA for SE Australia). First entries are for extremely wet seasons and second entries for extremely windy seasons. "$+/-$" symbols denote statistically significant anomalies; double symbols indicate that they are particularly large. Empty entries signify weak and/or statistically not significant anomalies.

|  | cyclone number | bomb number | min. SLP | deepening rate | lifetime | stationarity |
|---|---|---|---|---|---|---|
| California | $+/+$ | $/+$ | $--/--$ | $+/+$ |  |  |
| Central US |  |  |  |  |  |  |
| Florida | $++/++$ | $+/+$ |  |  |  | $++/++$ |
| Newfoundland | $+/$ | $+/+$ | $/-$ | $/+$ |  | $+/$ |
| Iceland | $++/-$ | $++/+$ | $-/-$ | $+/+$ |  |  |
| UK | $++/$ | $+/+$ | $-/-$ | $/++$ | $/+$ | $/-$ |
| Portugal | $++/+$ | $+/+$ | $--/--$ | $+/+$ | $+/+$ |  |
| Mediterranean | $++/+$ |  | $/-$ | $/+$ |  | $+/--$ |
| Germany | $+/$ | $/+$ | $/-$ | $/+$ |  | $/-$ |
| Japan | $+/-$ | $+/-$ |  |  |  | $+/$ |
| SE Australia | $+/$ |  | $/-$ |  |  |  |

lously many and stationary cyclones. During extremely wet summers, cyclones are anomalously frequent over Newfoundland, Iceland, Germany and particularly the UK and Japan, anomalously rare over the central US, anomalously deep over Iceland, the UK and southeastern Australia, anomalously long-lived over Iceland and the UK and anomalously stationary over California, the UK, the Mediterranean and Germany (Table 3). Extremely windy winters also exhibit strong spatial variability in the cy-

clone anomaly patterns that often differ from those of wet winters (Table 2). For instance, California is associated with positive anomalies in cyclone frequencies, bomb cyclone frequencies and deepening rates and strong negative anomalies in minimum SLP, while Florida is associated with positive anomalies in cyclone frequencies, bomb cyclone frequencies and stationarity. Windy winters over Iceland are characterised by anomalously few, but deep and strongly deepening cyclones, over Portugal, they are characterised by anomalously many, deep, strongly deepening and long-living cyclones, over southeastern Australia

by anomalously deep cyclones, and over Japan by anomalously few cyclones and bombs. During extremely windy summers, cyclones are anomalously frequent over the central US, the UK and Germany, anomalously rare over Portugal, anomalously deep over Newfoundland, Iceland, the UK and Japan, anomalously long-lived over California and Japan and anomalously stationary over California and the central US (Table 3).

Extremely dry and calm seasons also exhibit strong spatial variability in the cyclone frequencies and properties, but in many

regions of the world they are characterised by particularly few or weak cyclones that often originate further poleward and eastward (or from less far to the south and west) and have shorter lifetimes than in the climatology. Thus, also the absence of extratropical cyclones can contribute to the formation of extreme seasons. In a future study, it would be interesting to investigate globally whether calm and dry seasons are systematically associated with increased occurrences of other weather systems like,




**Table 3.** As Table 2 but for extreme seasons in local summer.

| | cyclone number | bomb number | min. SLP | deepening rate | lifetime | stationarity |
|---|---|---|---|---|---|---|
| California | | | | | /+ | +/++ |
| Central US | −/+ | | | | | −/+ |
| Florida | | | | | | |
| Newfoundland | +/ | | /− | /+ | | |
| Iceland | +/ | | −/− | +/ | +/ | |
| UK | ++/+ | | −/− | /+ | +/ | +/ |
| Portugal | /− | | | | | |
| Mediterranean | | | | | | +/ |
| Germany | +/+ | | | | | +/ |
| Japan | ++/ | | /− | /+ | /++ | |
| SE Australia | | | −/ | +/ | | |

for instance, surface anticyclones and blockings, which have been shown to be conducive to persistent dry episodes (e.g.,
Röthlisberger and Martius, 2019) or even entire dry seasons (e.g., Black et al., 2004).

Our results show that the presence or absence of extratropical cyclones and their characteristics like intensification and stationarity play a crucial role for the occurrence of extremely wet, dry, windy and calm seasons around the globe. While the relationship between extratropical cyclones and short-term wind and precipitation extremes is well-known (e.g., Gyakum, 1983; Wernli et al., 2002; Ulbrich et al., 2003; Pfahl and Wernli, 2012), for extremes on seasonal timescales their importance
has so far only been assessed for a small number of events (e.g., Davies, 2015; Flaounas et al., 2021; Boettcher et al., 2023). The systematic global analysis presented here provides unique insight into the cyclone characteristics during many different types of extreme seasons and highlights their seasonal and remarkably large regional variability (Tables 2 and 3). The results also indicate that changes in the number, geographical distribution and properties of extratropical cyclones with global warming would significantly affect extreme seasons around the globe. An essential next step is to evaluate such changes and their impact
on extreme seasons in future climate simulations.

As a caveat, it is noted that the study is based on output from a single climate model with relatively low spatial and temporal resolution. The complex and rich mesoscale substructures of surface precipitation and wind gusts are not adequately captured at this resolution. In addition, case studies have shown that longer-term climate conditions like anomalies in soil moisture, sea surface temperature, sea-ice extent or tropical convection can also contribute to the formation of extreme seasons (Fischer et al.,
2007; Dole et al., 2014; Hartmuth et al., 2022). This indicates that processes operating on different spatial and temporal scales are important for many extreme seasons around the globe. While the presence or absence of extratropical cyclones is crucial, it is not solely responsible for their occurrence. The exact interplay between short-term weather variability and longer-term climate conditions in creating extreme seasons remains to be investigated in future studies.

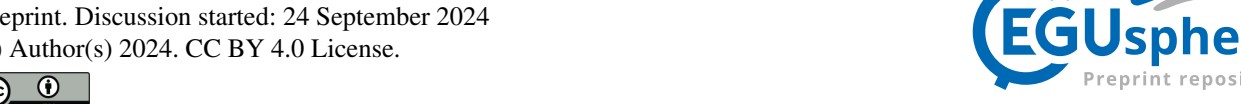

*Code and data availability.* The ERA5 extreme season objects can be accessed through the extreme seasons explorer (https://intexseas-
explorer.ethz.ch, last access: September 2024) and the ERA5 reanalyses from the ECMWF website
(https:// www.ecmwf.int/en/forecasts/datasets/reanalysis-datasets/era5, last access: September 2024). The CESM source code that was used
for the CESM-LE simulations is available from https://www.cesm.ucar.edu/models/cesm1.0/ (last access: May 2024). The model output of
the CESM-LE reruns, the CESM-LE extreme season objects and the cyclone data used in this study are available from the authors upon
request.

*Author contributions.* HB and HW designed the study, HB performed the study and wrote the manuscript, with feedback about the results
and text from HW.

*Competing interests.* HW is a member of the editorial board of Weather and Climate Dynamics. The authors have no other competing
interests to declare.

*Acknowledgements.* We thank Urs Beyerle (ETH Zurich) for performing the CESM-LE reruns, Michael Sprenger (ETH Zurich) for perform-
ing the cyclone tracking, and MeteoSwiss and ECMWF for granting access to the ERA5 reanalyses. We are grateful to Katharina Hartmuth
(ETH Zurich), Mauro Hermann and Matthias Röthlisberger (both formerly at ETH Zurich) for valuable comments and discussions. HB
received funding from the Swiss National Science Foundation (project 185049) and from the European Research Council H2020 research
and innovation program (INTEXseas, grant no. 787652).



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
