# Peer review of "Frequency anomalies and characteristics of extratropical cyclones during extremely wet, dry, windy and calm seasons in the extratropics"

_EGUsphere, 2024_

## Referee Comment (RC2)

*Review of manuscript egusphere-2024-2936 submitted to Weather and Climate Dynamics*

**Frequency anomalies and characteristics of extratropical cyclones during extremely wet, dry, windy and calm seasons in the extratropics**

by Hanin Binder and Heini Wernli

**General comments:**

This manuscript analyses the properties of extratropical cyclones during extreme seasons in both hemispheres using 1050 years of the CESM Large Ensemble historical simulations. Overall, this manuscript is written with an excellent level of detail and proficiency in English. I believe that this paper sheds light on the dynamics of extratropical cyclones and is suitable for the Weather and Climate Dynamics portfolio. I recommend that this paper be accepted, subject to minor comments.

**Specific comment:**

**1) Main comment on stationarity**: My primary comment is regarding the concept of cyclone stationarity. Stationarity is referenced multiple times, but a more detailed explanation of what defines stationarity is needed, as well as a discussion on the limitations of this definition.

2) **Line 274**: What is your hypothesis that stationary cyclones are more frequent at lower latitudes during the wet season?

3) **Line 343**: Could the positive anomaly be potentially linked to tropical cyclones in this region? (Gulf of Mexico and Central Pacific)

4) **Line 418**: What criteria were used to select these regions?

**Technical corrections:**

-Line 30: in winter 2013/14, the United Kingdom (UK).

-Line 121: obtain high-resolution three-dimensional model-level output. This needs more detail; for instance, how much has the vertical resolution increased? Can you describe previous and news vertical levels?

-Figure 1: The figure (i) description is not provided.

-Figure 3: Consider adding information about the contours in the caption or increasing their font thickness on the maps, as they are currently difficult to see.

-Line 301: In addition, anomalously stationary cyclones contribute to extremely wet summers over the exit regions of the NH storm tracks and large parts of central, eastern and southeastern Europe, the eastern Mediterranean and Kazakhstan (not shown). Could you include this in the supplementary material, at the very least?

-Line 361-364: Can you add a reference?

-Table 2: Adding lines to separate the rows would be helpful, as the lack of separation makes it difficult for the reader to follow.

---

## Author Response (AR1)

*Paper egusphere-2024-2936*

**Frequency anomalies and characteristics of extratropical cyclones during extremely wet, dry, windy and calm seasons in the extratropics**

by Hanin Binder and Heini Wernli

**Response to the reviewers' comments**

We thank the two reviewers for the constructive and thoughtful comments that helped us to improve our manuscript. Based on their suggestions, we have essentially made the following major changes:

- We have calculated the cyclone frequency anomalies and characteristics for all extreme season objects in ERA5 in the period 1950-2020 in the NH and SH extratropics, to investiagate systematically whether the results from the climate model agree with those from real-world extreme seasons (comment from reviewer 1). This analysis shows a good qualitative agreement between CESM and ERA5, which provides confidence in the findings from the climate model. We have included the figures for wet and windy DJF seasons in ERA5 in the supplementary material.
- We have shortened the abstract (comment from reviewer 1) and included a brief explanation of the definition and limitations of the stationarity measure used in our study (comment from reviewer 2).
- In the supplementary material, for the different types of extreme seasons during DJF and JJA we now include anomaly maps of the cyclone characteristics that are not shown in the main paper (comment from reviewer 2).

Below are the detailed replies to the points raised by the reviewers. Line numbers refer to the non-track-change version of the revised manuscript.

**Reviewer 1:**

*Summary*

*This paper investigates how extratropical cyclone occurrences influence extreme seasonal conditions using 1050 years of CESM-LE (large-ensemble) climate simulations and ERA5 reanalyses in present-day climate (1990-1999). It identifies patterns in extratropical cyclone frequency, intensity, and stationarity associated with extreme seasons across the Northern and Southern Hemispheres. Key findings include the link between increased extratropical cyclone frequency and intensity with wet and windy winters, while dry and calm seasons are marked by fewer or weaker cyclones. Regional variability is large, with the study highlighting shifts in storm tracks and distinct characteristics across different regions. The results highlight the importance of extratropical cyclones in shaping extreme seasonal weather and provide insights for future climate projections, although the authors note that the model's coarse resolution limits the capture of finer-scale features.*

*General Comments*

*I very much welcome this paper for publication given that it makes use of Large Ensemble climate simulations, which are critical to address uncertainty in historical and (for a potential future study) future projections. I also very much welcome the focus of the paper on extratropical cyclone associated extremes seasonality. Finally, I was very impressed by the scientific rigour and the level of detail of the scientific analysis.*

**Reply:** Many thanks for this positive assessment!

*The only major point I feel to flag is that I am not entirely convinced by how just looking at two (different for the type of extreme) seasons from ERA5 data can prove consistency with CESM-LE simulations, also considering the 2013/14 season is outside the CESM-LE sample years 1990-1999. I'd encourage the authors to better explain or perform a little further analysis in demonstrating how this link may be robust, for instance looking at more seasons from ERA5 data. I hypothesize this may be straightforward to perform, e.g. just re-use diagnostics tools already developed on easily downloadable era5 data, thus I suggest publication with minor revisions.*

**Reply:** Many thanks for this good suggestions. We agree that looking at only two case studies in ERA5 is not enough to prove consistency with the global results from CESM-LE. The main purpose of showing these well-known case studies is to illustrate the method. However, to investigate more systematically whether the CESM results are consistent with ERA5, we have calculated the cyclone frequency anomalies and characteristics for all ERA5 extreme season objects with an area larger than $10^5$ km$^2$ in the period 1950-2020 in the Northern and Southern Hemisphere extratropics and have produced global maps equivalent to those for CESM-LE in Section 4. Figures R1 and R2 show the anomaly patterns for extremely wet DJF seasons in ERA5, equivalent to Figs. 3 and 4 for CESM-LE in the manuscript. The fields look noisier because of the much smaller number of extreme seasons in ERA5 than in the 1050 year CESM ensemble. In total, there are 940 extreme season objects, and each grid point contains between 0 and 3 objects. The anomaly patterns look qualitatively very similar to those obtained in CESM-LE, exhibiting in many regions positive anomalies in the frequency of cyclones (Fig. R1a) and bombs (Fig. R1b), negative anomalies in minimum SLP (Fig. R1c) and positive anomalies in deepening rates (Fig. R1d). The cyclones often originate further equatorward and westward than climatologically (Fig. R2a,b), while the anomalies in lifetime and stationarity show relatively large spatial variability (Fig. R2c,d). The anomaly patterns of the 613 extremely windy DJF seasons in ERA5 are also mostly consistent with those in CESM-LE (Figs. R3 and R4): In particular in the main storm track regions windy seasons are associated with anomalously few, but intense, strongly intensifying, long-living, fast-moving and far southward and westward originating cyclones, while in oceanic regions at the edges of the storm tracks they are often associated with anomalously many and intense cyclones.

The systematic analysis of past extreme seasons in ERA5 confirms that CESM-LE is able to capture the frequency anomalies and characteristics of extratropical cyclones during extreme

seasons remarkably well, which justifies the use of the climate model and provides confidence in the findings of our study.

In the revised manuscript, we include Figs. R1-R4 in the supplementary material and briefly refer to them in the main paper in the new Section 3.2. We also include the equivalent figures for dry and calm DJF seasons in the supplement. Their anomaly patterns also qualitatively agree with those obtained in CESM-LE in most regions.

[Figure]

Fig. R1: As Fig. 3 in the main manuscript, but for extremely wet DJF seasons in ERA5. The shading shows seasonal anomalies with respect to the 1950-2020 winter climatology in the (a) number of cyclones, (b) number of bomb cyclones, (c) mean minimum SLP of the cyclones (hPa), and (d) mean deepening rate of the cyclones (Bergeron).

[Figure]

Fig. R2: As Fig. 4 in the main manuscript, but for extremely wet DJF seasons in ERA5. The shading shows seasonal anomalies in the cyclones' (a) genesis latitude (degrees), (b) genesis longitude (degrees), (c) lifetime (hours), and (d) stationarity (hours).

[Figure]

Fig. R3: As Fig. 7 in the main manuscript, but for extremely windy DJF seasons in ERA5. The same fields are shown as in Fig. R1.

[Figure]

Fig. R4: As Fig. 8 in the main manuscript, but for extremely windy DJF seasons in ERA5. The same fields are shown as in Fig. R2.

**Minor revisions**

*Abstract: I had a hard time following this long abstract. I think the paper would benefit greatly if the abstract could be made more concise and more tailored on key results.*

**Reply:** We agree that the original abstract is very long. We have shortened it in the revised version.

*Intro: If possible, please make a stronger link between how having a large ensemble of simulations helps/or has helped in previous studies to understand the extratropical cyclone extremes seasonality*

**Reply:** To the best of our knowledge, this is the first study that investigates the role of extratropical cyclones for extreme seasons using a large ensemble of climate simulations. The large ensembles are very useful for allowing statistically robust statements about anomaly patterns in frequency and intensity of extratropical cyclones during various types of extreme seasons throughout the extratropics. Such a global statistical characterisation would not be possible with the available reanalysis data alone, because at any location only very few such rare extreme seasons occurred in the last decades.

*Methods: Please explain better the cyclone detection algorithm and adaptation of extreme season detection.*

**Reply:** In the first paragraph of Section 2.2., we have included some additional sentences to better explain the cyclone detection algorithm. However, the algorithm is quite complex, and for a detailed description, the reader is referred to the study by Wernli and Schwierz (2006) and the supplementary material in Sprenger et al. (2017).

To identify extreme seasons, at each grid point we calculated seasonal mean values of precipitation and 10-m wind speed (in CESM) or 10-m wind gusts (in ERA5). In CESM, from the 1050 seasons we then selected the 25 seasons with the lowest values and the 25 seasonal with the highest values as extremely dry, calm, wet and windy seasons, respectively, which corresponds to a local return period of 42 years. In ERA5, because of the much smaller number of years, a statistical model has been fitted to the seasonal mean values to estimate the local return periods, and seasons with a return period of at least 40 years at the two tails of the distribution were considered to be extreme. In both CESM and ERA5, spatially coherent objects were then formed by connecting neighbouring grid points where the extreme values occurred in the same season.

The only differences between the methods used for ERA5 and CESM are therefore (i) the use of seasonal mean 10-m wind gusts in ERA5 vs. 10-m wind speeds in CESM, and (ii) the use of a statistical model to estimate local return periods to determine extreme seasons in ERA5 vs. the selection of the top 25 events with the largest values to determine extreme seasons in CESM.

In very similar words, these methods are described in the manuscript in the paragraph on lines 122-144. In the original version, it might have been confusing that we introduced the paragraph starting on line 122 by the sentence "To identify extreme (i.e, locally rare) seasons globally, we adopt a slightly modified version of the method developed by Röthlisberger et al. (2021) and Boettcher et al. (2023).", without making clear that the modifications were described further below. In the revised version, we have slightly rewritten this sentence.

*Results: The inclusion of the 2013/14 and 1988/89 seasons as real-world examples has the potential to strengthen the study. However, 2013/14 is outside the present-day CESM-LE sample, and they are just two. To better connect the statistical findings from the CESM analysis I suggest the authors to look at more ERA5 seasons, so as to create also a more coherent narrative, from which the paper would benefit.*

**Reply:** We agree, please see our reply to the general comment above.

*Summary: I find this sentence a bit vague at line 453-455 "The results also indicate that changes in the number, geographical distribution and properties of extratropical cyclones with global warming would significantly affect extreme seasons around the globe. An essential next step is to evaluate such changes and their impact on extreme seasons in future climate*

*simulations." More in details, I think it would be very valuable if the authors could add some insights on how the methodology contained in this paper could be used for warmer climate simulations, or even just analysing two different chunks of historical data, e.g. 1950-60 and 2010-2020, the latter already experiencing a warmer climate.*

**Reply**: The method used in this study for present-day climate simulations could directly be applied to future climate simulations, and then the changes between the present-day and the future climate could be investigated. Based on our findings for the present-day climate, which show the crucial role of extratropical cyclones for many extreme seasons around the world, we hypothesize that changes in the number of cyclones, their geographical distribution and their characteristics like intensity, intensification rate, and stationarity, could have a strong impact on the spatial distribution and the properties of extreme seasons. The sentence in the summary might seem a bit vague because we do not yet know the outcome of such a study (i.e., how the distribution and properties of extratropical cyclones will change in a future climate, and how this will affect the extreme seasons).

Thank you also for your suggestion to compare two different 10-year chunks of historical data to assess the role of climate change. Unfortunately, such an analysis is challenging due to the rareness of the extreme seasons. As we define extreme seasons based on a return period threshold of 40 years, at any location there are typically only around 1-2 extreme seasons in the 71 years of ERA5 data, and they do not necessarily occur in one of the specified 10-year chunks.

*When discussing the limitations related to the relatively coarse resolution of CESM-LE at line 456, the authors should highlight the trade-off between resolution and computational cost. Running CESM-LE with finer spatial resolution (e.g., 0.25° or less) would dramatically increase computational demands, potentially limiting the length or ensemble size of the simulations. Noting this trade-off would better contextualize the decision to use the current resolution and emphasize the value of leveraging large ensemble simulations for statistical robustness.*

**Reply:** We agree that the availability of large ensemble simulations, which allows for statistically robust statements, is most likely more valuable for our study than a much lower number of ensembles with higher spatial resolution. We now mention this on lines 465-468.

**Reviewer 2**

**General comments:**

*This manuscript analyses the properties of extratropical cyclones during extreme seasons in both hemispheres using 1050 years of the CESM Large Ensemble historical simulations. Overall, this manuscript is written with an excellent level of detail and proficiency in English. I believe that this paper sheds light on the dynamics of extratropical cyclones and is suitable*

*for the Weather and Climate Dynamics portfolio. I recommend that this paper be accepted, subject to minor comments.*

**Reply:** Many thanks for this positive general comment!

**Specific comment:**

*1) Main comment on stationarity: My primary comment is regarding the concept of cyclone stationarity. Stationarity is referenced multiple times, but a more detailed explanation of what defines stationarity is needed, as well as a discussion on the limitations of this definition.*

**Reply:** We define stationarity as the duration (in hours) that the centre of a cyclone remains within the area of the extreme season object. Fast-moving cyclones with a relatively direct and linear path spend less time within this area and are therefore less stationary compared to slow-moving cyclones or those with a more irregular path. Because for each object we apply the same method to both extreme seasons and the climatology, the resulting differences reveal whether cyclones moved faster or slower over the same area during extreme seasons compared to the climatological average, thereby providing meaningful insights into anomalies in stationarity for the specific extreme season.

The time a cyclone spends within the extreme season object also depends on the size and shape of the object itself. Our stationarity measure is suitable for the purpose of our analysis, where we are interested in the differences between extreme seasons and the climatology for the same object, but it would not be suitable for a comparison of absolute values among different objects (or, in that case, the duration spent within the extreme season object would somehow have to be weighted by its area).

In the revised manuscript, we added some sentences on lines 164-168 to briefly discuss the definition and limitations of the measure.

*2) Line 274: What is your hypothesis that stationary cyclones are more frequent at lower latitudes during the wet season?*

**Reply:** We hypothesize that during wet seasons, the subtropics are associated with anomalously many and anomalously stationary PV cut-offs and far equatorward-reaching stationary PV streamers, which are accompanied by anomalously stationary surface cyclones leading to prolonged precipitation over the same region and eventually extremely wet seasons. We discuss this briefly starting on line 279, and we have added an additonal sentence in the revised version.

*3) Line 343: Could the positive anomaly be potentially linked to tropical cyclones in this region? (Gulf of Mexico and Central Pacific)*

**Reply:** We think that the positive anomalies are indeed linked to extratropical cyclones, because the season shown here is DJF, when tropical cyclones rarely occur in the NH.

*4) **Line 418**: What criteria were used to select these regions?*

**Reply:** The regions were chosen subjectively, but with the main motivation to show the large spatial variability. In the revised version, we have included an additional entry over southern Chile and one over western Siberia, to increase the representation in the SH and over Asia.

**Technical corrections:**

*Line 30: in winter 2013/14**,** the United Kingdom (UK).*

**Reply:** Corrected, thank you.

*Line 121: obtain high-resolution three-dimensional model-level output. This needs more detail; for instance, how much has the vertical resolution increased? Can you describe previous and news vertical levels?*

**Reply:** The setup of our simulations and of the original CESM large ensemble simulations is essentially analogous, i.e., both have been run at a horizontal resolution of about 1° on 30 vertical levels. The key difference is that we store the output every 6 hours on all vertical model levels for all simulations, which we used in previous studies to identify various weather features. However, for the present study we actually do not need the output on all model levels, we only need surface-level data. To avoid confusion, in Section 2.1. we deleted the part of the sentence "to obtain high-resolution three-dimensional model-level output".

*Figure 1: The figure (i) description is not provided.*

**Reply:** Many thanks for spotting this typo. The label (h) should be written in front of "total duration of the cyclone tracks" and (i) in front of "stationarity". We have changed it accordingly.

*Figure 3: Consider adding information about the contours in the caption or increasing their font thickness on the maps, as they are currently difficult to see.*

**Reply:** Many thanks for pointing this out, the contour levels are indeed difficult to see in the figures. We have added information about them in the caption of Fig. 3.

*Line 301: In addition, anomalously stationary cyclones contribute to extremely wet summers over the exit regions of the NH storm tracks and large parts of central, eastern and southeastern Europe, the eastern Mediterranean and Kazakhstan (not shown). Could you include this in the supplementary material, at the very least?*

**Reply:** Thank you for this suggestion. We now include this figure in the supplementary material, together with the other cyclone characteristics that are not shown in the main paper for the different types of extreme seasons.

*Line 361-364: Can you add a reference?*

**Reply:** We have added some references.

*Table 2: Adding lines to separate the rows would be helpful, as the lack of separation makes it difficult for the reader to follow.*

**Reply:** Thank you for the good input, we have added a line between each row.